# DualFusion: Dual Adaptive Fusion for Multi-View Pedestrian Detection via View Reliability Modeling and Channel Reweighting

## Abstract

Multi-view pedestrian detection is crucial for safety-critical applications such as intelligent transportation and video surveillance, as it alleviates occlusions inherent in single-view perception. Recent methods project feature maps from multiple cameras onto a unified bird's-eye view (BEV), enabling spatially aligned cross-view fusion. However, most existing approaches fuse features uniformly, overlooking two critical aspects: (i) differences in view reliability (e.g., occlusion severity, projection distortion) and (ii) semantic correlations across feature channels (e.g., contours, foot-level cues). These issues lead to noisy aggregation and suboptimal detection accuracy. We propose DualFusion, a Dual Adaptive Fusion framework for robust BEV representation learning. It consists of two complementary modules: the Cross-View Feature Selector (CVFS), a Transformer-based encoder that performs spatially reliable, view-aware fusion, and the View-Channel Graph Attention (VCGA), which captures joint view–channel dependencies through context pooling and graph-inspired reweighting. Extensive experiments on public benchmarks show that DualFusion consistently outperforms state-of-the-art multi-view fusion approaches. Ablation studies further confirm the effectiveness of each component. Our approach provides a general paradigm for adaptive multi-view fusion, and code will be released upon acceptance.

## 1 Introduction

Pedestrian detection is a fundamental task in computer vision, with broad applications in surveillance Xu et al. (2022), autonomous driving Khan et al. (2023), and robotics (Mishra & Jabin, 2021). While single-view detectors have achieved notable progress (Falaschetti et al., 2024; Luo et al., 2024), occlusion remains a major challenge, as pedestrians are often only partially visible in a single view. Multi-view detection addresses this issue by using multiple cameras to observe the scene from different angles, based on the fact that a pedestrian occluded in one view may be visible in another, enabling complementary information fusion. Compared to single-view methods, multi-view systems fuse features into a shared bird's-eye view (BEV) (Alturki et al., 2025; Aung et al., 2025), improving spatial localization and robustness in cluttered environments (Chavdarova et al., 2018).

However, existing approaches typically use features in a uniform manner and suffer from two main limitations: (1) **Ignoring view reliability**. Most CNN-based fusion methods, such as MVDet (Hou et al., 2020), apply identical fusion weights across all views, regardless of occlusion or projection distortion. This makes it difficult to distinguish between informative and noisy observations, especially under occlusion, as illustrated in Fig. 1(a). (2) **Overlooking channel redundancy**. Most transformer-based methods, such as MVDeTr (Hou & Zheng, 2021), leverage deformable attention to flexibly sample spatial features across views. However, they lack explicit modeling of inter-view and inter-channel relationships, limiting their ability to suppress redundant activations and emphasize pedestrian-relevant semantic cues, as illustrated in Fig. 1(b). These limitations lead to sub-optimal BEV representations and hinder detection performance.

To address these limitations, we propose a Dual Adaptive Fusion framework (DualFusion) that improves BEV representations by performing view-aware spatial fusion and semantic channel recalibration, as illustrated in Fig. 1(c). This framework consists of two new modules, each designed to

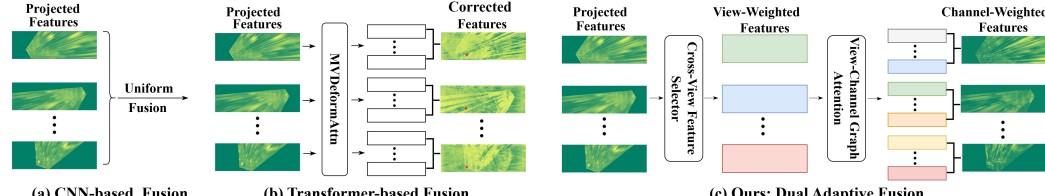

Figure 1: Comparison of multi-view pedestrian detection paradigms. (a) Fully convolutional methods (e.g., MVDet (Hou et al., 2020)) use uniform view fusion, ignoring view reliability and failing under severe occlusion. (b) Transformer-based methods (e.g., MVDeTr (Hou & Zheng, 2021)) flexibly aggregate spatial features, but lack explicit modeling of inter-view and inter-channel relationships. (c) Our proposed framework addresses both challenges via dual adaptive fusion: Cross-View Feature Selector (CVFS) assigns spatially adaptive weights across views (color intensity indicates view reliability), while View-Channel Graph Attention (VCGA) recalibrates semantic channels based on view-channel relevance (color intensity reflect channel importance).

tackle one identified limitation. Specifically, to address the view reliability issue, we introduce the **Cross-View Feature Selector (CVFS)** module, which employs a transformer encoder to perform global, view-aware feature fusion in the BEV space. By encoding position-aware tokens using multihead self-attention, CVFS adaptively aggregates complementary information from multiple views, effectively mitigating the effects of occlusion and projection inconsistency. To address the channel redundancy problem, we design the **View-Channel Graph Attention (VCGA)** module, which models joint view-channel dependencies through a combination of global and local pooling, followed by a graph-inspired multilayer perceptron (MLP). This enables adaptive channel-wise reweighting that highlights pedestrian-relevant semantic cues (e.g., feet and contours) while suppressing irrelevant noise, thereby enhancing the occupancy map quality for pedestrian localization.

The contributions of this paper are as follows:

- We propose a multi-view pedestrian detection framework that enhances BEV representations through dual adaptive fusion. The framework addresses two fundamental challenges in multi-view fusion: view reliability and channel redundancy.

- We introduce a Cross-View Feature Selector (CVFS) module that globally models interactions among projected multi-view features to learn spatially adaptive weights, enabling occlusion completion and suppression of unreliable observations.

- We propose a View-Channel Graph Attention (VCGA) module that captures joint viewchannel dependencies to enable fine-grained channel recalibration, enhancing pedestrianrelevant semantics while suppressing background noise.

- We demonstrate that both modules are plug-and-play and can be integrated into various multi-view detection methods. Experiments on the Wildtrack and MultiviewX benchmarks validate their effectiveness in improving detection performance across multiple baselines.

## 2 RELATED WORK

**Multi-view Pedestrian Detection**. Multi-view pedestrian detection Alturki et al. (2025); Aung et al. (2025) alleviates occlusion and visibility issues that hinder single-view detectors. By leveraging multiple calibrated cameras, it enables cross-view feature fusion to mitigate occlusion and projection distortion. Early approaches relied on probabilistic models (Coates & Ng, 2010; Sankaranarayanan et al., 2008) or graphical inference (Fleuret et al., 2007; Baqué et al., 2017), but suffered from high computational costs and limited scalability. Hou et al. (2020) proposed an end-to-end convolutional framework modeling spatial dependencies, while Hou & Zheng (2021) introduced deformable Transformers to better handle geometric misalignment. Subsequent works improved BEV feature quality via height-aware projection (Song et al., 2021; Qiu et al., 2022), frequency-domain fusion (Gao et al., 2022), ROI-based BEV refinement (Lee et al., 2023), and probabilistic occupancy volume Alturki et al. (2025). Vora et al. (2023) establish a benchmark and a minimal baseline robust

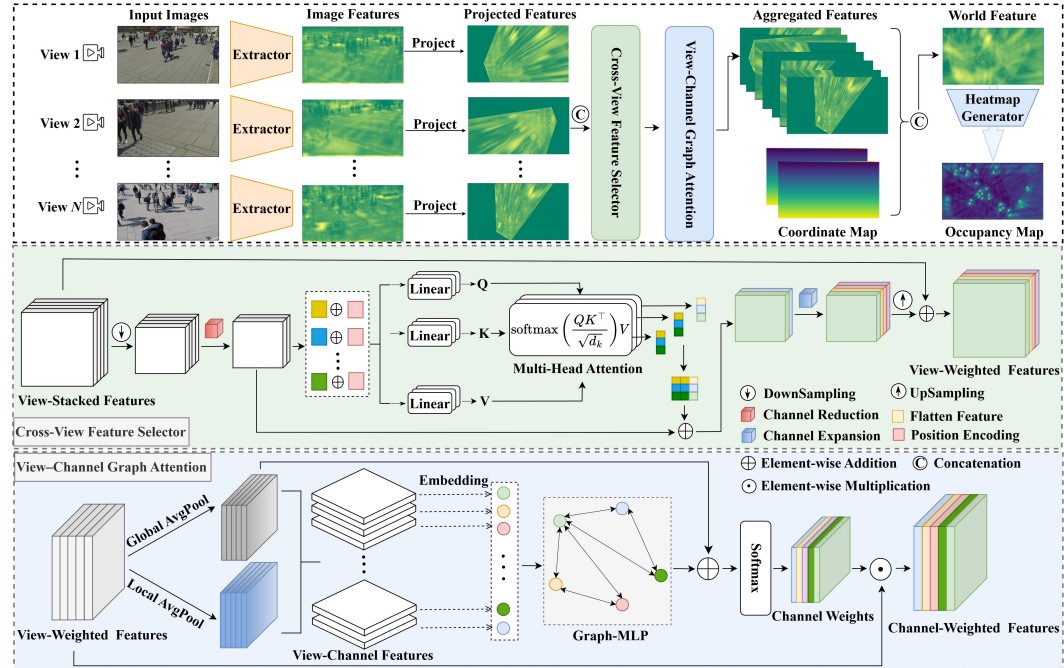

Figure 2: Overview of the proposed method (DualFusion). In the first stage, features from multiple cameras are projected onto the BEV plane and fused using a self-attention mechanism that adaptively weights contributions from different viewpoints (the green block corresponds to the Cross-View Feature Selector). In the second stage, the fused BEV features are transformed into a view–channel graph structure, where graph attention performs joint recalibration across both views and channels (the blue block corresponds to the View-Channel Graph Attention).

to varying camera counts/layouts and cross-scene transfer via ground-plane projection, view averaging, and dropview. Qiu et al. (2024) apply RoIAlign multi-view patches, fuse them in top view and regress occupancy, yielding stronger cross-dataset generalization than bottom-up BEV. However, these and later designs, such as supervised view-wise weighting Zhang et al. (2024a) and a foreground selector with vanilla channel attention Aung et al. (2024), largely use uniform fusion and overlook occlusion-aware view reliability and cross-view channel redundancy, motivating our Dual Adaptive Fusion.

**Transformer**. Transformer, initially developed for natural language processing (Vaswani et al., 2017), have been widely adopted in vision tasks for their ability to capture long-range dependencies (Wang et al., 2025; Shi, 2024). In multi-view detection, MVDeTr (Hou & Zheng, 2021) introduced deformable attention to sample informative spatial features under projection distortion. Inspired by these developments, our CVFS module adopts a Transformer encoder to perform global self-attention over concatenated multi-view BEV features, enabling dynamic, view-aware fusion.

**Channel and Graph Attention**. Channel attention mechanisms, such as Squeeze-and-Excitation (Hu et al., 2018), reweight feature channels using global pooling and lightweight MLPs to enhance informative responses. Variants (Gao et al., 2019; Lee et al., 2019; Qin et al., 2021) have further refined this idea for improved semantic sensitivity. Graph attention networks (Hu et al., 2021) model node interactions using learnable attention but incur high computational costs ($O(N^2)$), which limits scalability in multi-view scenarios. To address this, recent works (Tolstikhin et al., 2021; Liu et al., 2021; Hu et al., 2021) proposed MLP-based alternatives that approximate attention via global pooling and channel mixing, enabling efficient global reasoning. Motivated by these trends, VCGA approximates joint view-channel attention by combining global-local pooling and a graph-inspired MLP, achieving semantic channel reweighting with low computational overhead.

## 3 THE PROPOSED METHOD

We aim to improve multi-view pedestrian detection by enhancing the quality of BEV features. To this end, we address two key challenges in multi-view feature aggregation: (1) aligning spatial information across views under occlusion and distortion, and (2) selectively emphasizing semantically meaningful channels while suppressing noise.

To tackle these challenges, we propose two modules: CVFS and VCGA. CVFS enhances spatial alignment by assessing view reliability, while VCGA reduces channel redundancy through adaptive weighting based on global context and local activations. The overall architecture of the proposed DualFusion is shown in Fig. 2.

### 3.1 PROBLEM FORMULATION

The core objective of multi-view pedestrian detection is to project features from $N$ synchronized and calibrated cameras onto a common bird's-eye view (BEV) plane and to fuse them effectively. Given $N$ input images of shape $[3, H_i, W_i]$, we first employ a ResNet-18 backbone—shared across all views—to extract $C$-channel feature maps for each image. These maps are then resized to a unified spatial size $[H_f, W_f]$. Using the intrinsic matrix $A \in \mathbb{R}^{3 \times 3}$ and the extrinsic matrix $[R \mid t] \in \mathbb{R}^{3 \times 4}$, we compute the perspective transformation matrix $P_\theta$. Following the assumption in MVDet (Hou et al., 2020), all scene points are projected onto the ground plane ($z$=0). The image-space feature maps $F_s$ with pixel coordinates $(u, v)$ can thus be projected to ground-plane coordinates $(x, y)$, resulting in the warped BEV-aligned features $\widetilde{F_s}$, via:

$$\lambda \begin{pmatrix} u \\ v \\ 1 \end{pmatrix} = P_\theta \begin{pmatrix} x \\ y \\ z \\ 1 \end{pmatrix} = A\,[R \mid t] \begin{pmatrix} x \\ y \\ z \\ 1 \end{pmatrix}, \tag{1}$$

where $\lambda$ is a scale factor.

Through concatenating projected features with a coordinate map, we aggregate the ground plane feature map for the world feature. A heatmap generator $G_h$ is then used to incorporate local spatial context and produce an **occupancy map** for pedestrian localization. While multi-camera setups provide complementary observations, the key challenge remains: how to effectively fuse cross-view features to overcome occlusion, misalignment, and redundancy.

### 3.2 THE PROPOSED CROSS-VIEW FEATURE SELECTOR

Given projected BEV features of shape $[B, N, C, H_g, W_g]$, the Cross-View Feature Selector (CVFS) reshapes them so that, at each BEV grid cell $(h, w)$, features from all $N$ views are concatenated into a vector of dimension $N \cdot C$, forming a compact multi-view representation. A Transformer encoder with multi-head self-attention is then applied across BEV locations to aggregate these tokens, learning spatially adaptive weights that enhance reliable signals while suppressing occluded or noisy ones. The process consists of the following steps:

**1) Learnable camera embeddings.** To compensate for inherent viewpoint differences, we introduce a learnable bias vector $e_i \in \mathbb{R}^C$ for each camera and add it to the corresponding features:

$$\widetilde{\mathcal{F}}\,(b, i, c, u, v) = \mathcal{F}\,(b, i, c, u, v) + e_i\,(c)\,. \tag{2}$$

**2) Channel concatenation and spatial downsampling.** We concatenate the $N$ corrected view features along the channel dimension to form a tensor of shape $[B, N \cdot C, H_g, W_g]$, followed by a $3 \times 3$ convolution with stride $ds$ for spatial downsampling:

$$X_{b,k,u',v'} = \text{Conv}_{3 \times 3}^{\downarrow ds}\big(\widetilde{\mathcal{F}}\big) \;\in\; \mathbb{R}^{B \times N \cdot C \times H' \times W'}\,, \tag{3}$$

where $H' = H/ds$ and $W' = W/ds$.

**3) Channel projection.** The concatenated $N \cdot C$-Dimensional vector at each spatial location is projected into a more compact $D$-dimensional embedding via a $1 \times 1$ convolution:

$$Y(b, d, u', v') = \sum_{k=1}^{NC} W_{\text{in}}(d, k)\, X(b, k, u', v'), \tag{4}$$

where $d = 1, 2, \ldots, D$ and $W_{\text{in}} \in \mathbb{R}^{D \times (NC)}$ is a learnable weight matrix for channel reduction.

**4) Flattening and positional encoding.** The spatial grid of size $H' \times W'$ is flattened into a sequence of $S = H' \cdot W'$ tokens, each embedding fused multi-view information:

$$\mathbf{y}_s(b) = Y(b, :, u_s, v_s) \in \mathbb{R}^D, \quad s = 1, 2, \ldots, S \tag{5}$$

We then add 2D sine-cosine positional encoding $\text{PE}(u_s, v_s) \in \mathbb{R}^D$:

$$\mathbf{y}'_s(b) = \mathbf{y}_s(b) + \text{PE}(u_s, v_s) \in \mathbb{R}^D. \tag{6}$$

**5) Transformer self-attention fusion.** The position-encoded sequence is fed into a Transformer encoder to enable global, cross-view, and cross-channel interactions:

$$\mathbf{Z} = \text{TransformerEncoder}\big(\{\mathbf{y}'_s(b)\}_{s=1}^S\big) \in \mathbb{R}^{S \times B \times D}. \tag{7}$$

Here, attention weights act directly on the concatenated $[N \cdot C]$-D channel vector rather than per view separately, enabling the network to explicitly determine which views contribute most to pedestrian detection at each location.

**6) Spatial reprojection and channel recovery.** Encoded tokens are reassigned to their original spatial positions:

$$\hat{Y}(b, d, u', v') = Z(s, b, d) \quad ((u_s, v_s) \leftrightarrow s). \tag{8}$$

The original channel dimensionality is restored via another $1 \times 1$ convolution:

$$\hat{X}(b, k, u', v') = \sum_{d=1}^D W_{\text{out}}(k, d) \hat{Y}(b, d, u', v'), \tag{9}$$

where $k = 1, 2, \ldots, N \cdot C$, and $W_{\text{out}} \in \mathbb{R}^{(N \cdot C) \times D}$ is a learnable weight matrix for channel expansion.

**7) Spatial upsampling.** A transposed convolution is applied to upsample the feature map from the downsampled resolution $(H', W')$ back to the original BEV resolution $(H_g, W_g)$:

$$\widehat{\mathcal{F}} = \text{ConvTranspose}_{\uparrow ds}(\hat{X}) \in \mathbb{R}^{B \times (N \cdot C) \times H \times W}. \tag{10}$$

**8) Controlled residual fusion.** First, $\widehat{\mathcal{F}}$ is reshaped and restored to match the original multi-view feature dimensions, resulting in $\widehat{\mathcal{F}}'(b, i, c, u, v) \in \mathbb{R}^{B \times N \cdot C \times H \times W}$. Then a learnable scalar $\gamma \in R$ is subsequently introduced to balance the original and fused features via residual fusion:

$$\mathcal{F}'(b, i, c, u, v) = \widetilde{\mathcal{F}} + \gamma \widehat{\mathcal{F}}', \quad \gamma \in \mathbb{R}. \tag{11}$$

This residual gain mechanism allows the network to gradually emphasize attention-based corrections for occluded regions during training.

### 3.3 The proposed View-Channel Graph Attention

The VCGA module recalibrates multi-view BEV features by modeling joint dependencies across the view–channel dimension. Given the fused features from CVFS, $\mathcal{F} \in \mathbb{R}^{B \times N \times C \times H_g \times W_g}$, it applies graph-inspired attention to emphasize pedestrian-relevant semantics while suppressing redundant or noisy activations. The process consists of the following steps:

**1) View-channel graph construction.** We reshape the input tensor to $\mathbb{R}^{B \times (N \cdot C) \times H_g \times W_g}$, treating each of the $N \cdot C$ view-channel pairs as individual nodes in a conceptual graph. Each node is associated with a 2D feature map, while inter-node relationships are implicitly captured via subsequent attention modeling.

**2) Global and local pooling.** For each node, we compute a global context vector via global average pooling and a local descriptor over a $k \times k$ neighborhood:

$$\mathbf{g}_{i,c}^{(b)} = \frac{1}{HW} \sum_{u,v} \mathcal{F}(b, i, c, u, v), \quad \mathbf{l}_{i,c}^{(b)} = \frac{1}{|P|} \sum_{(u,v) \in P} \mathcal{F}(b, i, c, u, v), \tag{12}$$

where $P$ denotes the set of pixels within the local neighborhood centered at $(u, v)$.

**3) Node feature composition.** Each node's global and local descriptors are concatenated to form a unified feature representation:

$$\mathbf{y}_i^{(b)} = \left[\mathbf{g}_i^{(b)} \,\|\, \mathbf{l}_i^{(b)}\right] \in \mathbb{R}^{2C}. \tag{13}$$

**4) Graph-MLP aggregation.** All node features are concatenated into a single vector $\mathbf{y}^{(b)} \in \mathbb{R}^{2N \cdot C}$ and passed through a two-layer MLP:

$$\mathbf{z}^{(b)} = W_2 \cdot \sigma(W_1 \cdot \mathbf{y}^{(b)}), \tag{14}$$

where $W_1 \in \mathbb{R}^{\frac{2N \cdot C}{r} \times 2N \cdot C}$ and $W_2 \in \mathbb{R}^{N \cdot C \times \frac{2N \cdot C}{r}}$ are learnable parameters, $\sigma(\cdot)$ denotes the ReLU function, and $r$ is the reduction ratio. This MLP models global attention across all view-channel pairs without requiring an explicit adjacency matrix.

**5) Weight reconstruction and normalization.** The output is reshaped to per-node attention weights $\alpha_{i,c}^{(b)}$, which are combined with global descriptors and normalized:

$$s_{i,c}^{(b)} = \mathbf{g}_{i,c}^{(b)} + \gamma \cdot \alpha_{i,c}^{(b)}, \quad w_{i,c}^{(b)} = \sigma(s_{i,c}^{(b)}), \tag{15}$$

where $\gamma \in \mathbb{R}$ is a learnable scalar, and $\sigma$ denotes the sigmoid function.

**6) Channel recalibration.** Finally, the original BEV features are reweighted using the computed attention:

$$\widehat{\mathcal{F}}(b, i, c, u, v) = w_{i,c}^{(b)} \cdot \mathcal{F}(b, i, c, u, v). \tag{16}$$

This process adaptively emphasizes informative view-channel features, improving occupancy map quality for pedestrian localization while maintaining efficiency.

## 3.4 MODULE INTEGRATION STRATEGY

The proposed modules can be integrated with different baseline methods. As illustrated in Figure 2, our framework adopts a sequential strategy, applying CVFS before VCGA to process features extracted from the backbone. CVFS first restores occluded or unreliable spatial regions via cross-view fusion, while VCGA then performs joint view-channel recalibration to emphasize informative features. This ordering enables VCGA to operate on spatially completed representations, enhancing its ability to suppress noise and improve detection accuracy.

## 3.5 LOSS FUNCTION

Following MVDeTr (Hou & Zheng, 2021), we adopt a multi-branch supervision strategy to jointly optimize the global BEV prediction and the per-view image branches. The overall loss is defined as:

$$\mathcal{L}_{\text{total}} = \mathcal{L}_{\text{world}} + \alpha \cdot \frac{1}{N} \mathcal{L}_{\text{img}}, \tag{17}$$

where $\alpha$ is a balancing weight, and $N$ denotes the number of input camera views.

**Global BEV Loss.** The BEV prediction branch is supervised using a focal loss for occupancy map prediction and a masked L1 loss for offset regression:

$$\mathcal{L}_{\text{world}} = \mathcal{L}_{\text{focal}}(\hat{H}^w, H^w) + \mathcal{L}_{\text{reg}}(\hat{O}^w, O^w), \tag{18}$$

where $\hat{H}^w$ and $\hat{O}^w$ are the predicted occupancy map and offset field, while $H^w$ and $O^w$ are their corresponding ground-truth targets. The focal loss $\mathcal{L}_{\text{focal}}$ mitigates class imbalance, and the regression loss $\mathcal{L}_{\text{reg}}$ is computed only at ground-truth object centers.

**Image-View Loss.** For each input view $v = 1, 2, \cdots, N$, we apply auxiliary supervision to enhance the view-specific branches. Each view is supervised on image feature map prediction, offset regression, and object size estimation:

$$\mathcal{L}_{\text{img}} = \sum_{v=1}^{N} \left(\mathcal{L}_{\text{focal}}(\hat{H}^v, H^v) + \mathcal{L}_{\text{reg}}(\hat{O}^v, O^v) + \lambda_{\text{wh}} \cdot \mathcal{L}_{\text{reg}}(\hat{W}^v, W^v)\right), \tag{19}$$

where $\hat{H}^v$, $\hat{O}^v$, and $\hat{W}^v$ denote the predicted image feature, offset, and object size maps for view $v$, and $H^v$, $O^v$, $W^v$ are their ground-truth counterparts. The hyperparameter $\lambda_{\text{wh}}$ balances the object size regression term.

## 4 EXPERIMENTS

The proposed method was evaluated through a series of experiments and compared against existing multi-view pedestrian detection methods.

### 4.1 IMPLEMENTATION DETAILS

Our method is implemented based on MVDetr (Hou & Zheng, 2021), where the proposed dual adaptive attention modules are inserted after the per-view feature projection onto the world (BEV) plane. Specifically, we use the Adam optimizer with a learning rate of $5 \times 10^{-4}$ and train the model with a batch size of 1 on a single NVIDIA RTX 4090 GPU. To evaluate the generalization capability of our module, we further integrate it into two representative frameworks—MVDet (Hou et al., 2020) and SHOT (Song et al., 2021)—using the same sequential integration strategy (i.e., CVFS followed by VCGA) as in MVDetr, and assess their performance through ablation studies. These auxiliary implementations follow their original training protocols and loss functions.

### 4.2 DATASETS

We evaluate the proposed method on two public datasets:

**Wildtrack (Chavdarova et al., 2018)**. The Wildtrack dataset is a real-world dataset covering a 12 m × 36 m area captured by seven synchronized cameras. Each image is $1080 \times 1920$ pixels. The persons per frame (ppf) of this dataset is 20. Wildtrack comprises 400 frames; following the MVDet split (Hou et al., 2020), we use the first 360 frames for training and the remaining 40 for testing.

**MultiviewX (Hou et al., 2020)**. The MultiviewX dataset is a synthetic dataset rendered with the Unity engine over a 16 m × 25 m urban plaza. Compared to Wildtrack, MultiviewX exhibits nearly twice the pedestrian density (ppf is 40) to offset the limited appearance variability inherent in synthetic data. The images are also captured at $1080 \times 1920$ resolution. This dataset likewise contains 400 frames, with the final 40 frames reserved for evaluation.

### 4.3 EVALUATION METRICS

Following existing studies (Hou et al., 2020), we adopt four evaluation metrics: Multiple Object Detection Accuracy (MODA), Multiple Object Detection Precision (MODP), precision, and recall. Rather than relying on bounding box overlaps, we evaluate performance using predicted ground-plane occupancy maps. A detection is considered a true positive if the Euclidean distance between the predicted location and the ground-truth foot point is within a predefined threshold. Consistent with prior work, we apply non-maximum suppression (NMS) at a spatial resolution of 0.5 meters. Among these metrics, we use MODA as the primary one, as it penalizes both false positives and false negatives, offering a balanced assessment of detection quality.

### 4.4 COMPARISON WITH EXISTING METHODS

**Quantitative comparisons**. We compare our method with **14 existing approaches**, including EarlyBird (Teepe et al., 2024a), TMVD (Qiu et al., 2024), BoosterShot (Hwang et al., 2024), M-MVOT (Zhang et al., 2024b), TrackTacul (Teepe et al., 2024b), GMVD (Vora et al., 2023), Exploiting Key Points (Gao et al., 2022), 3DROM (Qiu et al., 2022), MVDetr (Hou & Zheng, 2021), SHOT (Song et al., 2021), MVDet (Hou et al., 2020), Deep Occlusion (Baqué et al., 2017), DeepMCD (Chavdarova & Fleuret, 2017), RCNN-2D/3D (Xu et al., 2016). Table 2 presents the results on the WildTrack and MultiviewX datasets in terms of MODA, MODP, precision, and recall. Overall, DualFusion is highly competitive on both benchmarks. On WildTrack, it attains the best MODA and best precision, and is second in recall. This pattern matches our design: VCGA suppresses background responses, yielding fewer false positives and thus higher precision and MODA; meanwhile, CVFS performs cross-view completion, recovering many occluded pedestrians and keeping recall high. On MultiviewX, DualFusion achieves the best MODP, second-best precision, and third-best MODA. The MODP gain is consistent with the sharper, more localized peaks produced by CVFS (better spatial localization on the BEV grid), while the strong precision again benefits from VCGA's view–channel recalibration.

Table 1: Comparison with recent methods on public benchmarks. **Bold** and underlined denote the optimal and suboptimal.

| Method | WildTrack | | | | MultiviewX | | | |
|---|---|---|---|---|---|---|---|---|
| | MODA | MODP | Prec | Recall | MODA | MODP | Prec | Recall |
| RCNN-2D/3D (Xu et al., 2016) | 11.3 | 18.4 | 68.0 | 43.0 | 18.7 | 46.4 | 63.5 | 43.9 |
| DeepMCD (Chavdarova & Fleuret, 2017) | 67.8 | 64.2 | 85.0 | 82.0 | 70.0 | 73.0 | 85.7 | 83.3 |
| Deep Occlusion (Baqué et al., 2017) | 74.1 | 53.8 | 95.0 | 80.0 | 75.2 | 54.7 | 97.8 | 80.2 |
| MVDet (Hou et al., 2020) | 88.2 | 75.7 | 94.7 | 93.6 | 83.9 | 79.6 | 96.8 | 86.7 |
| SHOT (Song et al., 2021) | 90.2 | 76.5 | 96.1 | 94.0 | 88.3 | 82.0 | 96.6 | 91.5 |
| MVDetr (Hou & Zheng, 2021) | 91.5 | 82.1 | **97.4** | 94.0 | 93.7 | 91.3 | **99.5** | 94.2 |
| 3DROM (Qiu et al., 2022) | 93.5 | 75.9 | 97.2 | 96.2 | 95.0 | 84.9 | 99.0 | 96.1 |
| Exploiting key points (Gao et al., 2022) | 92.4 | 82.4 | 97.3 | 95.4 | 93.9 | 91.8 | 99.4 | 94.7 |
| GMVD (Vora et al., 2023) | 85.4 | 76.7 | 95.2 | 89.9 | 86.9 | 79.8 | 97.2 | 89.6 |
| EarlyBird (Teepe et al., 2024a) | 91.2 | 81.8 | 94.9 | 96.3 | 94.2 | 90.1 | 98.6 | 95.7 |
| TMVD (Qiu et al., 2024) | 89.4 | 73.8 | 95.9 | 93.4 | 89.8 | 80.6 | 97.1 | 92.6 |
| BoosterShot (Hwang et al., 2024) | 92.9 | **82.6** | 96.5 | 96.3 | 94.2 | 91.9 | **99.5** | 94.6 |
| M-MVOT (Zhang et al., 2024b) | 92.1 | 81.3 | 94.5 | **97.8** | **96.7** | 86.1 | 98.8 | **97.9** |
| TrackTacul (Teepe et al., 2024b) | 93.2 | 77.5 | 97.3 | 95.8 | 96.1 | 90.4 | 99.0 | 97.1 |
| **DualFusion (ours)** | **93.8** | 81.5 | **97.4** | 96.3 | 95.5 | **92.3** | 99.4 | 96.0 |

**Qualitative comparisons: occupancy maps**. As shown in Fig. 3, our occupancy maps (middle) consistently outperform MVDetr (left). In red-boxed regions where the baseline shows weak or missing responses, our method produces strong peaks aligned with GT, achieving better recall under occlusion and weak-view conditions. Shadow- and projection-induced artifacts as well as false activations in empty areas are reduced, yielding a cleaner background. Pedestrian clusters appear sharper, more complete, and better separated. These gains come from the synergy of our modules: CVFS enhances cross-view completion and spatial focusing, while VCGA recalibrates view–channel pairs to emphasize pedestrian-consistent channels and suppress noise-sensitive ones. The resulting energy maps are sparser and higher-contrast, improving both recall and precision and enhancing detection clarity.

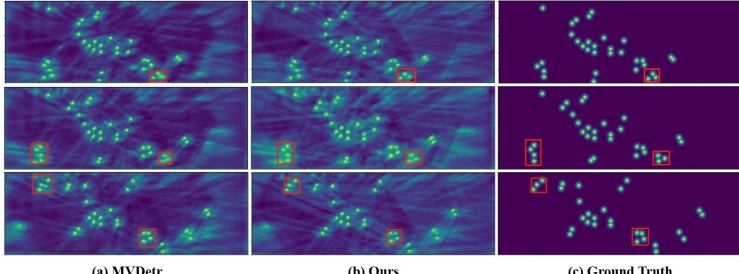

(a) MVDetr      (b) Ours      (c) Ground Truth

Figure 3: Comparisons of occupancy maps. Each point in the ground truth represents a pedestrian in the BEV. Ours (middle) yields sharper, higher-contrast occupancy peaks than MVDetr (left), aligning better with GT (right). Gains stem from CVFS (cross-view completion and spatial focusing) and VCGA (view–channel recalibration), which suppress noise and emphasize pedestrian-consistent evidence.

**Qualitative comparisons: projected features**. Fig. 4 summarizes inputs, projections, and module effects across (a–e). (a) shows an occluded, shadowed view; (b) its BEV projection, where boundary striping is a projection artifact. (c) MVDetr yields diffuse activations: shadows/empty regions get high background responses while foothold peaks are weak—showing limited cross-view completion and noise suppression. (d) Our method sharpens, localizes, and suppresses peripheral responses; CVFS linearly mixes views per BEV cell and applies spatial self-attention to route non-local evidence, filling occlusions and down-weighting unreliable regions/artifacts. (e) the channel-aggregation map emphasizes pedestrian-related structures and de-emphasizes shadows/stripes via global–local pooling and a graph-inspired MLP that models view–channel pairs. In short, CVFS decides where to trust (cross-view completion and spatial focusing) and VCGA decides what to

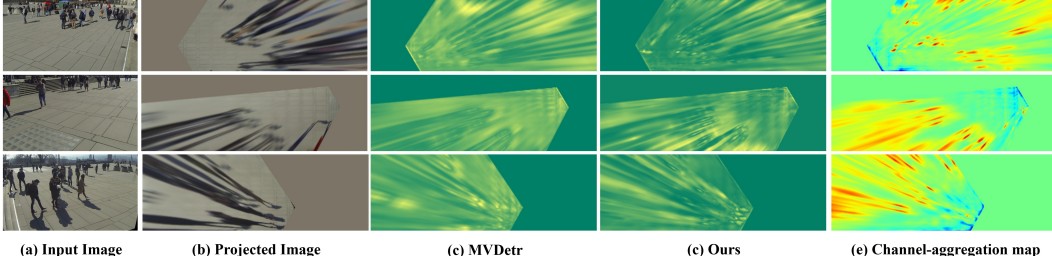

| (a) Input Image | (b) Projected Image | (c) MVDetr | (c) Ours | (e) Channel-aggregation map |
|---|---|---|---|---|

Figure 4: Comparisons of projected features. (a) original images from cameras; (b) projected images; (c) projected features of MVDetr; (d) projected features of our method: sharper, more localized responses at pedestrian footholds with peripheral/irrelevant areas suppressed; (e) channel-aggregation map: visualized by projecting learned channel weights onto the projected features as a heatmap; high weights (warm colors) concentrate on pedestrian-related regions and extension directions, whereas shadows/stripe artifacts and peripheral background are down-weighted (cool colors).

trust (channel-wise denoising/selection); remaining fine stripes stem from projection geometry, not model failure.

### 4.5 ABLATION STUDIES

**Effects of CVFS and VCGA modules**.  Table 2 shows the effect of the proposed CVFS and VCGA modules on the MVDetr baseline. CVFS performs dynamic cross-view completion and, when used alone, significantly improves detection performance, highlighting the effectiveness of spatial self-attention in handling occlusions. VCGA focuses on view–channel recalibration, and the results indicate that channel-wise selection enhances localization accuracy. When combined, the two modules provide further benefits, leading to substantial improvements in both detection coverage and precision—validating the effectiveness of the overall design. Additional analyses, including channel map comparisons, baseline model effects, module integration strategies, and window size sensitivity, are provided in the **Appendix**.

Table 2: Effects of CVFS and VCGA modules on detection performance.

| Modules | | Metrics | | | |
|---|---|---|---|---|---|
| CVFS | VCGA | MODA | MODP | Precision | Recall |
| X | X | 91.5 | **82.1** | **97.4** | 94.0 |
| ✓ | X | 92.9 | 80.8 | 97.3 | 95.5 |
| X | ✓ | 92.8 | 81.4 | **97.4** | 95.3 |
| ✓ | ✓ | **93.8** | 81.5 | **97.4** | **96.3** |

## 5 CONCLUSIONS

This paper proposes a multi-view pedestrian detection framework that integrates a Cross-View Feature Selector (CVFS) and a View-Channel Graph Attention (VCGA) module to enhance BEV feature quality via spatial fusion and adaptive channel reweighting. CVFS completes occluded features and suppresses irrelevant context, while VCGA highlights key semantic channels and filters redundancy. Experiments on the WildTrack and MultiviewX datasets show that the proposed method consistently outperforms state-of-the-art approaches. Ablation studies further validate the effectiveness and complementarity of the two modules in handling complex occlusion scenarios. Future work will focus on further enhancing the lightweight design of our modules and improving generalization and ensuring adaptability across cross-scenario settings.

## ETHICS STATEMENT

The authors affirm that this study adheres to the ICLR Code of Ethics.

## REPRODUCIBILITY STATEMENT

The authors ensure the reproducibility of this work. Implementation details are provided in Section 4.1, the datasets used in this study are publicly available (see Section 4.2), and the code will be released upon acceptance.

## THE USE OF LARGE LANGUAGE MODELS (LLMS)

ChatGPT-5 was used solely for language polishing and grammar checking in some parts of this paper. The following prompt was employed: *"Please check language errors in these texts. Do not change the content if there are no language errors."* No content or scientific contributions were generated by the model.

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

# A  APPENDIX

**Qualitative comparisons: channel maps**.   As shown in Fig. 5(a), the baseline world feature exhibits uniformly moderate activations across channels and views, indicating limited selectivity. In contrast, Fig. 5(b) shows that VCGA assigns differentiated weights, highlighting informative channel-view pairs. After weighting (Fig. 5(c)), activations become more focused and sparse, suggesting improved feature discrimination and noise suppression. This focused activation pattern confirms that VCGA adaptively emphasizes pedestrian-related, cross-view-consistent channels while suppressing redundant responses, serving as quantitative evidence for the qualitative channel-aggregation visualization shown above.

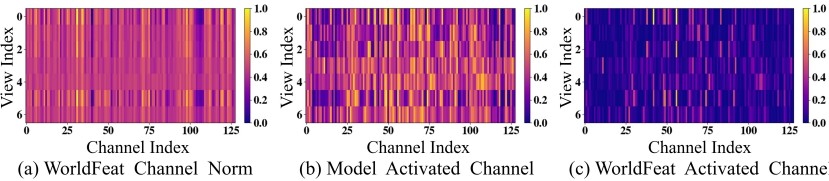

Figure 5: Comparison of three channel maps: (a) Baseline world feature activations, averaged spatially and normalized per channel; (b) View-channel attention weights from VCGA, normalized for visualization; (c) Weighted activations after applying VCGA via element-wise multiplication.

**Effects of baseline models**. As mentioned earlier, the proposed modules can be integrated into various baseline models. Table 3 presents the results of incorporating our modules into MVDet, SHOT, and MVDetr. On the WildTrack dataset, our method improves MODA by 1.8% for MVDet, 1.3% for SHOT, and 2.3% for MVDetr. Similarly, on the MultiviewX dataset, it yields gains of 8.9%, 5.1%, and 1.8% respectively. These consistent improvements demonstrate the general applicability and effectiveness of our modules in enhancing detection performance across different frameworks.

Table 3: Comparisons of integrating the proposed module into various baseline models.

| Method | WildTrack | | | | MultiviewX | | | |
|---|---|---|---|---|---|---|---|---|
| | moda | modp | prec | recall | moda | modp | prec | recall |
| MVDet | 88.2 | 75.7 | 94.7 | 93.6 | 83.9 | 79.6 | 96.8 | 86.7 |
| SHOT | 90.2 | 76.5 | 96.1 | 94.0 | 88.3 | 82.0 | 96.6 | 91.5 |
| MVDetr | 91.5 | **82.1** | **97.4** | 94.0 | 93.7 | 91.3 | **99.5** | 94.2 |
| MVDet+Ours | 90.0 | 81.1 | 96.5 | 92.2 | 92.8 | 91.9 | 98.9 | 93.8 |
| SHOT+Ours | 91.5 | 80.0 | 95.9 | 95.6 | 93.4 | 91.0 | 98.7 | 94.6 |
| MVDetr+Ours | **93.8** | 81.5 | **97.4** | **96.3** | **95.5** | **92.3** | 99.4 | **96.0** |

**Effects of module integration strategies**. From the results in Table 4, it is evident that applying channel-wise graph attention before cross-view fusion fails to fully leverage the complementary information across views. In contrast, performing spatial-level dynamic cross-view completion via CVFS first, followed by channel–view joint recalibration with VCGA, leads to the most effective improvement in detection coverage and overall localization accuracy. The parallel configuration achieves the highest MODP and precision, suggesting it effectively combines the strengths of both modules—part of the features are enhanced by CVFS, while others are finely filtered by VCGA. Their combined outputs facilitate more accurate positive sample identification. Among the three configurations, the sequential pipeline of cross-view fusion followed by channel–view attention recalibration proves to be optimal, as it both recovers occluded information and suppresses noise and redundancy across channel–view dimensions, ultimately achieving the highest MODA and recall.

Table 4: Comparison of different integration strategies for CVFS and VCGA to identify the most effective architectural composition.

| Method | MODA | MODP | Precision | Recall |
|---|---|---|---|---|
| VCGA → CVFS | 92.6 | 81.1 | 97.3 | 95.3 |
| CVFS → VCGA | **93.8** | 81.5 | 97.4 | **96.3** |
| CVFS // VCGA | 93.4 | **81.7** | **97.8** | 95.5 |

**Effects of window size in local pooling**. We further investigate the impact of the local pooling window size $K$ on the performance of the VCGA module, as shown in Table 5. When $K=1$, pooling degenerates to global averaging, capturing only cross-view global statistics and missing fine-grained details between neighboring grids. Increasing to $K=3$ introduces moderate local context that helps suppress small-region noise, but MODP and Recall remain close to $K=1$. At $K=5$, the window strikes a balance between global context and local focus, enhancing sensitivity to local objects while resisting background noise and yielding the best overall gains in detection accuracy and coverage. With $K=7$, MODP peaks (more precise localization), but MODA and Precision drop and the Recall increase comes with more false positives, indicating that overly large windows introduce irrelevant background. Therefore, we set $K=5$ in our experiments.

Table 5: Effects of local pooling window size in VCGA.

| Pooling Size | MODA | MODP | Precision | Recall |
|---|---|---|---|---|
| $K=1$ | 92.8 | 81.8 | 96.9 | 95.8 |
| $K=3$ | 93.0 | 81.0 | **97.4** | 95.5 |
| $K=5$ | **93.8** | 81.5 | **97.4** | **96.3** |
| $K=7$ | 92.8 | **82.4** | 96.4 | **96.3** |

