# OpenReview forum: "DualFusion: Dual Adaptive Fusion for Multi-View Pedestrian Detection via View Reliability Modeling and Channel Reweighting"
_ICLR.cc/2026/Conference — ICLR 2026 Conference Withdrawn Submission_

### Official Review · Reviewer_ffUz · 2025-10-27

**Soundness:** 3
**Presentation:** 2
**Contribution:** 3
**Rating:** 4
**Confidence:** 4

**Summary:**

The paper introduces DualFusion, a framework for robust multi-view pedestrian detection in safety-critical tasks like transportation and surveillance. Traditional BEV-based fusion methods combine multi-camera features uniformly, ignoring differences in view reliability (due to occlusions or distortions) and semantic correlations across feature channels.

DualFusion addresses these issues with two modules:
1. Cross-View Feature Selector (CVFS) – a Transformer-based encoder that performs spatially reliable, view-aware fusion
2. View-Channel Graph Attention (VCGA) – a graph-inspired module that models joint dependencies across views and feature channels.

Experiments on public benchmarks show consistent performance gains over existing methods, and ablation studies validate each module’s contribution. DualFusion thus establishes a general paradigm for adaptive multi-view fusion in BEV-based perception.

**Strengths:**

1. The paper is well-organized. It describes the two proposed modules, the cross-view feature selector (CVFS) and the view-channel graph attention (VCGA) modules, in detail. The proposed modules are easy to reproduce.

2. The proposed framework combines spatial and channel-wise attention mechanisms to automatically re-weight different areas in feature maps by the proposed modules.

3. The paper includes comprehensive quantitative experiments to demonstrate that the proposed framework is competitive to the state-of-the-art methods.

**Weaknesses:**

1. The novelty of the cross-view feature selector is limited. The proposed cross-view feature selector module only leverages the multi-head self-attention layers in the transformer blocks to emphasize important and suppress unimportant features in feature maps without futher modifying the architecture of the self-attention layers for better attentive feature generation.

2. The idea of the view-channel graph attention is pretty interesting, but the purpose of using graph attention is still vague. It is better to explicitly clarify why the graph attention outperforms conventional channel-wise attention mechanisms.

3. The proposed modules and the framework only achieve competitively quantitative results compared to the state-of-the-art methods. Based on the experimentation, the proposed framework does not fully resolve the issues mentioned in the paper.

**Questions:**

No further questions.

---

### Official Review · Reviewer_J2Yv · 2025-10-30

**Soundness:** 3
**Presentation:** 3
**Contribution:** 2
**Rating:** 4
**Confidence:** 4

**Summary:**

This paper proposes two new modules for soft view selection and GNN-based channel reweighting on BEV space. Experiments demonstrate its effectiveness on two benchmark multi-view detection datasets.

**Strengths:**

1. The proposed methods got SOTA results on Wildtrack.
2. The VCGA module incorporates the channel attention that is overlooked by previous multi-view detection methods.

**Weaknesses:**

The main weaknesses lie in the module novelty and performance:

1. The CVFS module is overly simplistic, being essentially just a standard attention operation across different views.

2. Compared to MVDetr, the design of CVFS is much weaker, since MVDetr not only applies attention across multiple views but also takes projection geometry into account.

4. The performance improvement observed by inserting CVFS into MVDetr may in part stem from an effect similar to adding more attention layers to the backbone. Therefore, simply deepening the original MVDetr architecture might also achieve comparable results.

5. As in the main result table, the model performance is not SOTA on the MultiviewX dataset, which does not convince us of the effectiveness of the proposed module on different scenarios.

6. The experiment is not comprehensive. More experiments should be conducted on larger datasets, such as CVCS, for a more complete comparison, since Wildtrack and MultiviewX are too small, only containing 360 frames for training and 40 for testing.

**Questions:**

As above.

---

### Official Review · Reviewer_kWJf · 2025-10-30

**Soundness:** 2
**Presentation:** 2
**Contribution:** 1
**Rating:** 2
**Confidence:** 5

**Summary:**

This paper proposes DualFusion method with channel-wise and view-wise feature fusion for multi-view pedestrian detection task, which separates the classic unified self-attention into two folds. In details, Cross-View Feature Selector (CVFS) adopts multi-head self-attention to predict view-wise weights for feature selection. View-Channel Graph Attention (VCGA) utilizes Graph-MLP to obtain channel-wise weights as importance of different feature channels. Experiments are conducted on mainstream datasets Wildtrack and MultiviewX.

**Strengths:**

1. The differences among CNN-based Fusion, Transformer-based Fusion, and proposed DualFusion methods are fully discussed.

2. Recent state-of-the-arts are evaluated to compare performance on mainstream datasets.

**Weaknesses:**

1. Performance: DualFusion is not state-of-the-art on various metrics, i.e., MODP and Recall on Wildtrack, as well as MODA, Precision, and Recall on MultiviewX, which cannot support the claimed advantages of DualFusion over CNN-based and Transformer-based Fusion.

2. Interpretability: Without any extra supervision, how can CVFS and VCGA capture the correct view-wise and channel-wise features? Can they learn the geometric rules, e.g., near views contribute more to detect one pedestrian, and far views focus less? And what does each channel represent like “contours, foot-level cues” mentioned in Abstract?

3. Confusing Visualizations: Figure 3 attempts to show the results between proposed DualFusion and baseline MVDetr, but there are no significant differences observed. The “irrelevant area” is also hard to be connect with the corresponding parts in Figure 4.  And the highlights before and after channel attention is also hard to interpret in Figure 5. The connection of Graph-MLP is similar to the weight of a vanilla MLP-based channel attention.

4. Related Works and Motivation: the cited methods in Section “Channel and Graph Attention” are too outdated in 2018-2021. In Section “Multi-view Pedestrian Detection”, “Qiu et al (2024), Zhang et al (2024a) and Aung et al (2024)” are defined as “overlook occlusion-aware view reliability and cross-view channel redundancy”, but the proposed DualFusion does not adopt explicit occlusion handling, like extra supervision or segmentation masks, and the sparsity of channel activation is also not explicitly constrained. These are conflict with the motivation.

5. Experiments: Now that Aung et al (2024) has already employed “vanilla channel attention”, and Zhang et al (2024a) also investigate supervised weight fusion, why they are not compared in state-of-the-art comparison in Table 1 or evaluated as similar modules with CVFS and VCGA in ablation study? And why only classic baselines MVDet and MVDetr are adopted? Now that there are more recent Transformer-based and CNN-based methods in Table 1.

Typo: There are two “(c)” in Figure 4.

**Questions:**

Refer to the Weaknesses section

---

### Note · Authors · 2025-11-14

I have read and agree with the venue's withdrawal policy on behalf of myself and my co-authors.